**Data Availability Statement:** These analyses were performed using data from the Corporate Data Warehouse domains that are available only within a secure research environment behind the US

# Understanding clinician connections to inform efforts to promote high-quality inflammatory bowel disease care

Shirley Cohen-Mekelburg[1,2]*, Tony Van[1], Xianshi Yu[3], Deena Kelly Costa[4,5], Milisa Manojlovich[4,5], Sameer Saini[1,2], Heather Gilmartin[6], Andrew J. Admon[7,8], Ken Resnicow[9], Peter D. R. Higgins[2], Geoffrey Siwo[2], Ji Zhu[3], Akbar K. Waljee[1,2]

**1** VA Center for Clinical Management Research, LTC Charles Kettles VA Medical Center, Ann Arbor, Michigan, United States of America, **2** Division of Gastroenterology & Hepatology, University of Michigan Medicine, Ann Arbor, Michigan, United States of America, **3** Department of Statistics, University of Michigan Medicine, Ann Arbor, Michigan, United States of America, **4** School of Nursing, Yale University, New Haven, Connecticut, United States of America, **5** Section on Pulmonary, Critical Care & Sleep Medicine, Department of Internal Medicine, Yale University, New Haven, Connecticut, United States of America, **6** Denver/Seattle Center of Innovation, VA Eastern Colorado Healthcare System, Aurora, Colorado, United States of America, **7** Division of Pulmonary and Critical Care Medicine, Department of Internal Medicine, University of Michigan Medicine, Ann Arbor, Michigan, United States of America, **8** Pulmonary Service, LTC Charles Kettles VA Medical Center, Ann Arbor, Michigan, United States of America, **9** Department of Health Education and Health Behavior, University of Michigan School of Public Health, Ann Arbor, Michigan, United States of America

* shcohen@umich.edu

## Abstract

### Background

Highly connected individuals disseminate information effectively within their social network. To apply this concept to inflammatory bowel disease (IBD) care and lay the foundation for network interventions to disseminate high-quality treatment, we assessed the need for improving the IBD practices of highly connected clinicians. We aimed to examine whether highly connected clinicians who treat IBD patients were more likely to provide high-quality treatment than less connected clinicians.

### Methods

We used network analysis to examine connections among clinicians who shared patients with IBD in the Veterans Health Administration between 2015–2018. We created a network comprised of clinicians connected by shared patients. We quantified clinician connections using degree centrality (number of clinicians with whom a clinician shares patients), closeness centrality (reach via shared contacts to other clinicians), and betweenness centrality (degree to which a clinician connects clinicians not otherwise connected). Using weighted linear regression, we examined associations between each measure of connection and two IBD quality indicators: low prolonged steroids use, and high steroid-sparing therapy use.

Department of Veterans Affairs firewall. In order to comply with VA privacy and data security policies and regulatory constraints, only aggregate summary statistics and results of our analyses are permitted to be shared for publication. The authors have provided detailed results of the analyses in the paper. These restrictions are in place in order to maintain Veteran privacy and confidentiality. Access to these data can be granted to persons who are not employees of the VA; however, there is an official protocol that must be followed. The authors also confirm that VA policies are currently being developed that should allow an interested researcher to obtain a de-identified, raw dataset upon request with a data use agreement. Those wishing to access the data that were used for this analysis may contact Jennifer Burns, MHSA, who is a senior data manager at the VA Center for Clinical Management Research, to discuss the details of the VA data access approval process. Her contact information is as follows: Email: Jennifer. Burns@va.gov. Address: UM North Campus Research Complex, Department of Veterans Affairs, 2800 Plymouth Road Bldg 16, Ann Arbor, MI, 48109-2800. She has access to the data, knowledge of the current and evolving VA policies for sharing data and is included on the IRB for this study. The corresponding author, Shirley Cohen-Mekelburg (shcohen@med.umich.edu) is also available to discuss the details of the VA data access approval process.

**Funding:** This work was supported by the National Institute of Health through the Michigan Institute for Clinical and Health Research (KL2TR002241 to S.C.M.), VA Health Services Research And Development Service of the VA Office of Research and Development (1IK2HX002587-01A1 to H.G). The views expressed in this article are those of the authors and do not necessarily represent the views of the Department of Veterans Affairs or the National Institute of Health. The funders had no role in study design, data collection and analysis, decision to publish, or preparation of the manuscript. The authors have no other conflicts of interest to disclose.

**Competing interests:** The authors have declared that no competing interests exist.

## Results

We identified 62,971 patients with IBD and linked them to 1,655 gastroenterologists and 7,852 primary care providers. Clinicians with more connections (degree) were more likely to exhibit high-quality treatment (less prolonged steroids beta -0.0268, 95% CI -0.0427, -0.0110, more steroid-sparing therapy beta 0.0967, 95%CI 0.0128, 0.1805). Clinicians who connect otherwise unconnected clinicians (betweenness) displayed more prolonged steroids use (beta 0.0003, 95%CI 0.0001, 0.0006). The presence of variation is more relevant than its magnitude.

## Conclusions

Clinicians with a high number of connections provided more high-quality IBD treatments than less connected clinicians, and may be well-positioned for interventions to disseminate high-quality IBD care. However, clinicians who connect clinicians who are otherwise unconnected are more likely to display low-quality IBD treatment. Efforts to improve their quality are needed prior to leveraging their position to disseminate high-quality care.

## Background

Inflammatory bowel disease (IBD) is a chronic inflammatory condition of the gastrointestinal tract that affects an estimated 3 million Americans and often leads to disability and a low quality of life. Scientific advancements have revolutionized IBD treatment with the development of several novel and effective medications. Despite this, patients with IBD continue to suffer from high rates of suboptimal disease control, preventable disease complications, and disability [1–4]. These negative health outcomes result in part from variation in clinical practice, which is influenced by patient, clinician, and organizational factors. One example of this variation in IBD care is the common use of prolonged steroid treatment in some facilities and by some clinicians, and the underutilization of steroid-sparing therapies [5, 6]. There is an urgent need to understand how to better optimize the dissemination and adoption of high-quality clinical practices to minimize variation in high-quality treatment and deliver better IBD care.

Collaboration and exchange of information between individuals facilitate the spread of knowledge and adoption of beneficial practices. This is a central tenet of social network theory, which emphasizes the importance of connections. In social network theory, individuals take action based on their network's environment and an individual's positioning in a network influences their behavior [7]. These concepts have been applied to a wide variety of health areas. Highly connected individuals have been shown to disseminate information and increase the effectiveness of public health interventions in producing widespread behavioral change [8]. One classic example is that of a randomized controlled study of public health interventions across Honduran villages that compared an intervention strategy targeting highly connected individuals nominated by others as compared to a non-targeted strategy. The interventions included multivitamins for micronutrient deficiencies, and chlorine for water purification. The study found that the strategy of targeting highly connected individuals was more effective than a non-targeted strategy in disseminating these high-quality public health practices [8]. Our overarching goal is to apply this knowledge from the public health literature to facilitate the dissemination of high-quality IBD clinical practices among clinicians in a health system network.

Collaborative networks are bounded by a defined group of individuals who work together to achieve a common purpose. Often a clinician's network is influenced by their years of experience, specialized training, and health system factors such as facility referral practices or dedicated IBD-specialized clinics. Within clinician networks, some individuals are characterized by their many acquaintances, while others have few acquaintances but play a critical role in connecting key people who may not otherwise interact. In healthcare, a gastroenterologist who has a large referral practice and shares several patients with each referring clinician would be considered a highly connected member of their network. Based on social network theory, this gastroenterologist would be well-positioned to disseminate high-quality clinical practices within their network through patient encounters, informal communication with other clinicians, and perhaps even more formalized continuing medical education programs with their colleagues. On the other hand, a less connected gastroenterologist with a smaller group of referring clinicians with whom they share patients may be less efficient at disseminating high-quality clinical practices.

Current organizational efforts to promote high-quality IBD clinical practices, including IBD-specialized medical home models, learning health systems, clinician education, and shared decision making initiatives are not designed to target specific clinicians or groups within healthcare networks [9–11]. Studies have demonstrated that higher patient volume and more specialized IBD training are associated with improved outcomes [12–14]. Nonetheless, access to high volume IBD centers and IBD-specialized clinicians is limited. Clinician training and experience aside, designing interventions to target highly connected clinicians in a network, could facilitate the wide dissemination and adoption of high-quality IBD practices. However, we have little understanding of clinicians who provide IBD care, their connections with other clinicians, nor the quality of their IBD clinical practices. Our study objective is to examine whether highly connected clinicians who treat patients with IBD are more likely to provide high-quality IBD care than less connected clinicians. This study will identify the potential need for improving the practices of highly connected clinicians, laying the foundation for developing interventions that leverage clinician networks to disseminate high-quality IBD care.

## Methods

### Study population

The VHA is the largest integrated health system within the United States. The VHA provides an excellent model for studying variation in and dissemination of high-value clinical practices in IBD given its collaborative network of medical centers. Further, the VHA prioritizes high-value care, and VHA care is less affected by common external barriers to care such as variation in insurance coverage of evidence-based IBD-targeted therapies that can hinder high-value care in the community. We identified patients with IBD who were managed within the Veterans Health Administration (VHA) between January 1, 2015, and December 31, 2018. Patients with IBD were identified using a combination of International Classification of Diseases, Tenth Revision (ICD-10) codes for ulcerative colitis and Crohn's disease that have been validated in VHA data [15]. We identified clinicians (physicians, nurse practitioners, and physician assistants) who cared for patients with IBD linked by office visit encounters. In the VHA, primary care providers act as the central coordinator of care, with gastroenterologists manage IBD-specific medical decision making. Therefore, all gastroenterology providers and primary care providers who cared for an IBD patient through at least one office visit were included. Other specialties were excluded as they are less likely to be accountable for chronic IBD care. There were no additional exclusions.

## Study outcomes

There were two primary outcomes representative of high-quality IBD treatment: clinicians with fewer patients who are prolonged steroid users and clinicians with a higher percentage of previous or current prolonged steroids users receiving steroid-sparing therapy. Both outcomes are considered quality metrics in IBD care and recommendations are to minimize use of prolonged steroids and maximize steroid-sparing therapy with immunomodulators or biologics in this context, though threshold benchmarks do not currently exist [9, 16]. Prior data suggests that there is wide variation in the use of prolonged steroids and the use of steroid-sparing therapies [5, 17, 18]. These measures directly relate to the ability to achieve steroid-free remission and prevents IBD-related complications, which is why we chose them as our outcome measures. Prolonged steroid use was defined on a patient-level by two or more corticosteroid prescriptions (budesonide, cortisone, hydrocortisone, prednisone, prednisolone, or methylprednisolone) with at least a 14 day supply dispensed within 90 days of each other, based on previously published work [5]. Steroid-sparing therapy was defined by prescription of an immunomodulator (azathioprine, 6-mercaptopurine, methotrexate) and/or a prescription of a biologic or small molecule approved for the treatment of IBD (infliximab, adalimumab, certolizumab, golimumab, vedolizumab, ustekinumab, or tofacitinib). These variables were transformed to clinician-level measures and reported as percentage of a clinician's patients who achieved each endpoint [16]. All clinicians who cared for a patient were considered equally accountable for their care.

## Network construction

Clinicians who cared for patients with IBD during the study period were linked based on previously described methodology, and a patient-sharing network was created [19]. A network is constructed from *nodes* representing unique clinicians, and *edges* or connections, which are represented by the patients that two clinicians share. Given the natural tendency for most care to occur within a given facility, the facility was considered the network boundary.

## Derivation of measures of connection

We measured clinician connections by quantifying three network variables (Fig 1): degree centrality, closeness centrality, and betweenness centrality.

*Degree centrality* characterizes connections and is defined by the number of clinicians with whom a single clinician shares patients. These connections provide a foundation for interaction, communication, collaboration, and information sharing. Degree centrality is measured as a count of the number of patient-sharing connections with other clinicians. For example, a clinician who shares patients with six other clinicians, would have a degree of 6 (Fig 1). Operationally, a clinician with a high degree centrality is more connected than a clinician with a lower degree centrality. Measuring the degree centrality within a network may identify clinicians who are well positioned to disseminate evidence-based practices to a higher number of clinicians with whom they share patients.

*Closeness centrality* characterizes a clinician's reach (via shared clinician contacts) to all other clinicians in a network. It is measured using geodesic distance. Two clinicians who are directly connected by shared patients are half as distant as two clinicians who are indirectly connected through a third clinician with whom they each share patients. The graph distance between any two clinicians 'a' and 'b' equals one plus the smallest number of clinicians through which they are connected. We denote this distance by $d(a, b)$. The closeness centrality of clinician 'a' is calculated using their graph distance to all clinicians in their

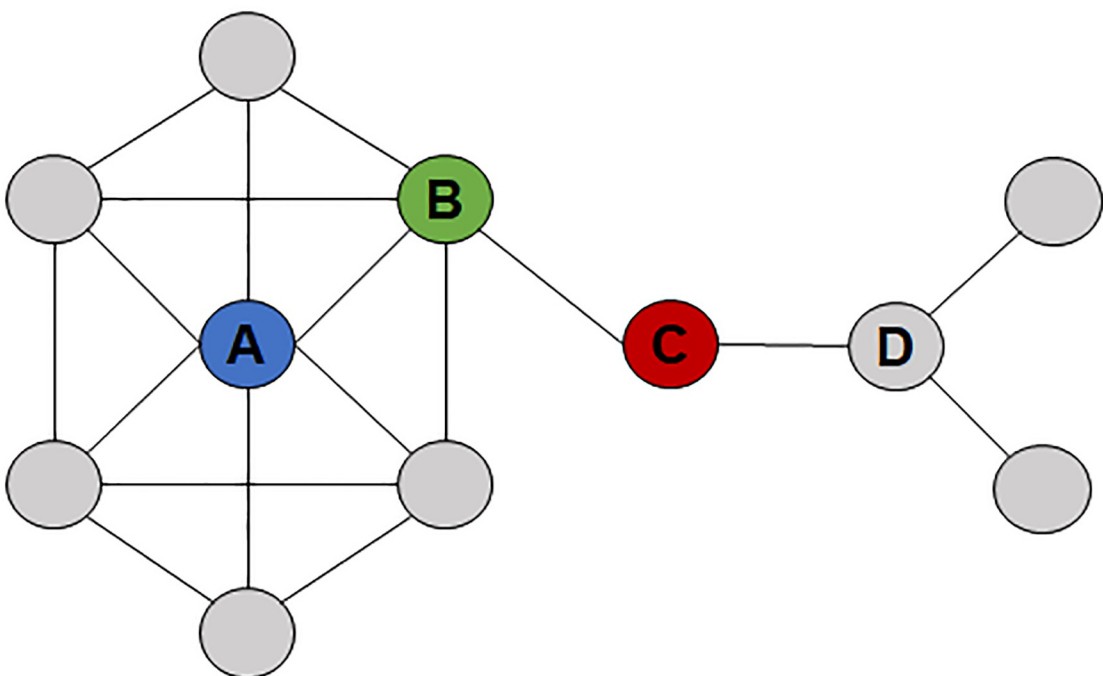

**Fig 1. Visualization of three centrality measures: A network with 11 nodes (gray circles; unique providers) who are connected by 18 edges (black lines; the patients they share).** Provider A has the highest degree in the network (degree = 6). Provider C has the highest betweenness as C is positioned on the shortest path between any two nodes respectively to the left and right of C. Provider B has the highest closeness with the shortest distance to all other providers in the network.

facility, annotated as:

$$\sum_b \frac{1}{d(a, b)}$$

where 'b' represents the total number of clinicians in a network within the reach of clinician 'a'. For example, a clinician with connections to many people at the shortest distance will have a higher closeness than a clinician with fewer, more distant connections (Fig 1). The value and interpretation of closeness may vary when considering a clinician's position within a specified facility as compared to the entire study population, as closeness is dependent on the total number of clinicians in a network. Further, normalization is required to allow for comparison of closeness between networks of varying sizes. Operationally, a clinician with a higher closeness who has a farther reach via patient-sharing contacts is more connected than a clinician with lower closeness who has a much shorter reach. This creates an opportunity to disseminate evidence-based practices more widely not only to the clinicians with whom they share patients, but clinicians they are able to reach through shared clinician contacts.

*Betweenness centrality* characterizes the degree to which a clinician connects clinicians not otherwise connected. A clinician with high betweenness possesses a critical role in connecting clinicians within a network, as in their absence, connections between other clinicians would be less likely. It is measured by the shortest path between two clinicians in a network (Fig 1). We use $g_{ab}$ to denote the number of shortest paths between a pair of clinicians 'a' and 'b' and use $g_{ab}(c)$ to denote the number of the shortest paths that pass-through clinician 'c'. The

betweenness centrality of 'c' is then defined by

$$\sum_{ab} \frac{g_{ab}(c)}{g_{ab}},$$

where the summation is made over all clinician pairs ('a', 'b') in the facility. Operationally, a clinician with a high betweenness does not necessarily have a high number of clinician contacts but has an important role in sharing patients with clinicians who might otherwise be isolated from other clinicians. This creates an opportunity to disseminate evidence-based practices to clinicians with few clinician contacts.

## Clinician-related covariates

We collected data on clinician type (physician, nurse practitioner, physician assistant) and specialty (gastroenterology, primary care). Clinician connections do not specifically describe an individual clinician's "collaborative spirit," but rather their role in a clinical network, which may be a function of their clinical practice setting. Therefore, we also collected clinical practice characteristics such as volume of patients with IBD, number of visits for patients with IBD, the proportion of patients with Crohn's disease and ulcerative colitis; and clinical practice characteristics such as location (urban or rural), region, and facility complexity designation [20].

## Statistical analysis

All continuous variables were reported using means and standard deviations, and categorical variables were reported using frequency and percentages. We assessed the bivariate relationship between clinician- and clinical practice-related characteristics and each measure of collaboration using one-way ANOVA. Continuous variables were categorized into quartiles to facilitate comparisons by differences in centrality. All p-values <0.05 were considered statistically significant. We used a multivariable linear regression to examine the association between each measure of centrality and clinician-level outcome measures. We controlled *a priori* for specialty, clinician type, practice location, facility complexity designation, and volume of patients with IBD. Since the dependent variables are percentages derived from the proportion of a clinicians' patients achieving a specified outcome, the variances of the dependent variable are not homogeneous and are larger for clinicians with fewer patients. Therefore, they do not meet the assumption of homoscedasticity associated with linear regression models. To account for this heteroscedasticity, the regression was weighted by clinicians' IBD patient volume. The weighted linear regression allows clinicians with a smaller IBD patient volume to contribute less than clinicians with a larger IBD volume.

## Sensitivity analyses

In the VHA, primary care providers act as the central coordinator of care, and in many contexts will actively participate in IBD management, including evaluating IBD exacerbations and prescribing IBD-targeted therapy. However, in primary care providers and gastroenterologists also may play different roles in IBD care and therefore differentially influence IBD patient outcomes. The role of primary care providers in IBD care may vary by health system, whereas gastroenterologists more consistently manage IBD-specific medical decision making. Therefore, we conducted a subgroup analysis examining the association between each measure of centrality and the primary outcomes for only gastroenterology providers. All analyses were performed in R version 4.0.5. This study was approved by the VA Ann Arbor Institutional Review Board (IRB# IRB 2019–1204). Consent was waived given the use of deidentified data.

## Results

We identified 62,971 patients with IBD and linked them to 1,655 gastroenterologists and 7,852 primary care providers, with which they had a total of 1,041,588 office visits between 2015 and 2018 across 130 facilities nationwide. These clinicians were predominantly physicians (n = 6,867, 72.2%) rather than nurse practitioners (n = 1,954, 20.5%) or physician assistants (n = 686, 7.2%), in a primary care provider role (n = 7,852, 82.6%), and cared for a mean of 16 (standard deviation [sd] 26) IBD patients over the study period. There were 1,655 gastroenterologists (17.4%) included in the study who cared for a mean of 37 (sd 54) IBD patients over the study period. Overall, most clinicians cared for patients in an urban setting (n = 8,707, 92.6%), and in affiliation with a facility with a high complexity designation (n = 7,719, 80.9%) (Table 1). Clinician-level outcomes varied. On average, 6.9% (sd 12.6%) of clinicians' IBD patients were prolonged steroid users, and 25.6% (sd 37.4%) of these clinicians' prolonged steroid users received steroid-sparing therapy with an immunomodulator or biologic.

Clinicians' connections varied, with an overall mean degree centrality of 9.5 (sd 13.4), mean closeness centrality of 39.8 (sd 23.0), and mean betweenness centrality of 61.1 (sd 287.9) (Fig 2). Visually, this variation is apparent within facilities (Fig 3). An association exists between clinicians' level of centrality in their network and clinician-related characteristics. Gastroenterologists were more likely to be highly connected than primary care providers, with a higher mean degree centrality (gastroenterologist 23.19 [sd 25.48] vs. primary care providers 6.60 [sd 5.76], p < 0.001), a higher mean closeness centrality (gastroenterologist 51.42 [sd 23.87] vs. primary care providers 37.37 [sd 22.13], p<0.001), and higher mean betweenness centrality (gastroenterologist 230.79 [sd 616.55] vs. primary care providers 25.36 [sd 113.65], p<0.001). Physician assistants were similarly more likely to be highly connected (mean degree 10.75 [sd 15.11]; p = 0.038, mean closeness 40.70 [sd 24.60]; p = 0.002, and mean betweenness 85.16 [sd 351.05]; p = 0.076) than physicians (mean degree 9.38 [sd 13.52], mean closeness 40.18 [sd 22.98], and mean betweenness 59.27 [sd 300.98]) or nurse practitioners (mean degree 9.43 [sd 12.36], mean closeness 38.18 [sd 22.74], and mean betweenness 59.18 [sd 203.65]) (Table 1). Clinicians with a higher IBD volume and more IBD visit encounters were also more connected than those with lower IBD volumes and fewer visit encounters. Clinicians with the top quartile of IBD patient volume demonstrated a mean degree centrality of 21.88 (sd 21.43, p<0.001), mean closeness centrality of 52.40 (sd 23.57, p<0.001), and mean betweenness centrality of 193.79 (sd 541.20, p<0.001) as compared to clinicians at the bottom quartile of IBD patient volume with a mean degree centrality 2.55 (sd 2.37), mean closeness centrality 28.54 (21.22), and mean betweenness centrality 3.73 (sd 21.09). Clinicians with the top quartile of IBD visit volume demonstrated a mean degree centrality of 20.01 (sd 21.97, p<0.001), mean closeness centrality of 50.33 (sd 24.07, p<0.001), and mean betweenness centrality of 180.54 (sd 537.83, p<0.001) as compared to clinicians at the bottom quartile of IBD patient volume with a mean degree centrality 2.74 (sd 2.60), mean closeness centrality 28.82 (21.15), and mean betweenness centrality 4.39 (sd 23.12).

Clinicians' practice settings were also associated with their level of connection. Clinicians who practiced in an urban setting were more connected (mean degree centrality 9.83 [sd 13.84, p<0.001], mean closeness centrality 41.63 [sd 22.95, p<0.001], and mean betweenness centrality 63.09 [sd 298.82, p = 0.033]) than those who practiced in a rural setting (mean degree centrality 5.91 [sd 6.54], mean closeness centrality 18.50 [sd 11.72], mean betweenness centrality 33.68 [sd 97.71]). Finally, clinicians who were affiliated with facilities with the highest complexity designation were more connected (mean degree centrality 11.69 [sd 16.43; p<0.001], mean closeness centrality 50.33 [sd 19.53; p<0.001]) than those affiliated with facilities with the lowest complexity designation (mean degree centrality 5.27 [sd 5.29], mean closeness centrality

**Table 1. Measures of centrality and associated provider characteristics.**

| Provider characteristic | Total | Degree (mean, sd) | P-value | Closeness (mean, sd) | P-value | Betweenness (mean, sd) | P-value |
|---|---|---|---|---|---|---|---|
| **Total** | 9507 | 9.5 (13.4) | | 39.8 (23.0) | | 61.1 (287.9) | |
| Specialty | | | <0.001 | | <0.001 | | <0.001 |
| Gastroenterology | 1655 (17.41%) | 23.19 (25.48) | | 51.42 (23.87) | | 230.79 (616.55) | |
| Primary care | 7852 (82.59%) | 6.6 (5.76) | | 37.36 (22.13) | | 25.36 (113.65) | |
| Type of provider | | | 0.038 | | 0.002 | | 0.076 |
| Nurse practitioner | 1954 (20.55%) | 9.43 (12.36) | | 38.18 (22.74) | | 59.18 (203.65) | |
| Physician | 6867 (72.23%) | 9.38 (13.52) | | 40.18 (22.98) | | 59.27 (300.98) | |
| Physician assistant | 686 (7.22%) | 10.75 (15.11) | | 40.70 (24.6) | | 85.16 (351.05) | |
| Number of IBD patients over study period | | | <0.001 | | <0.001 | | <0.001 |
| < 4 | 2377 (25%) | 2.55 (2.37) | | 28.54 (21.22) | | 3.73 (21.09) | |
| < 8 | 2377 (25%) | 5.47 (3.62) | | 36.89 (19.86) | | 16.71 (50.54) | |
| < 13 | 2377 (25%) | 8.08 (4.74) | | 41.41 (20.79) | | 30.32 (109.49) | |
| >= 13 | 2376 (24.99%) | 21.88 (21.43) | | 52.4 (23.57) | | 193.79 (541.2) | |
| Number of IBD visits over study period | | | <0.001 | | <0.001 | | <0.001 |
| < 6 | 2377 (25%) | 2.74 (2.6) | | 28.82 (21.15) | | 4.39 (23.12) | |
| < 16 | 2377 (25%) | 6.1 (4.6) | | 37.6 (20.57) | | 20.22 (88.03) | |
| < 31 | 2377 (25%) | 9.13 (6.46) | | 42.5 (20.87) | | 39.39 (120.52) | |
| >= 31 | 2376 (24.99%) | 20.01 (21.97) | | 50.33 (24.07) | | 180.54 (537.83) | |
| Proportion of IBD patients with Crohn's | | | <0.001 | | <0.001 | | <0.001 |
| < 20% | 2377 (25%) | 5.23 (5.49) | | 33.72 (21.96) | | 17.57 (52.24) | |
| < 33.3% | 2377 (25%) | 10.99 (12.26) | | 43.37 (22.6) | | 78.14 (349.28) | |
| < 50% | 2377 (25%) | 15.5 (20.48) | | 45.59 (23.52) | | 119.68 (405.55) | |
| >= 50% | 2376 (24.99%) | 6.24 (7.29) | | 36.56 (22.11) | | 29.08 (189.09) | |
| Proportion of IBD patients with UC | | | <0.001 | | <0.001 | | <0.001 |
| < 44.4% | 2377 (25%) | 8.43 (12.65) | | 38.68 (22.84) | | 52.9 (308.97) | |
| < 57.1% | 2377 (25%) | 13.53 (19.17) | | 43.8 (23.87) | | 99.97 (334.29) | |
| < 71.4% | 2377 (25%) | 10.71 (11.26) | | 43.36 (22.18) | | 72.91 (343.51) | |
| >= 71.4% | 2376 (24.99%) | 5.3 (5.4) | | 33.41 (21.78) | | 18.69 (53.43) | |
| Location of practice | | | <0.001 | | <0.001 | | 0.033 |
| Urban | 8707 (92.64%) | 9.83 (13.84) | | 41.63 (22.95) | | 63.09 (298.82) | |
| Rural | 687 (7.31%) | 5.91 (6.54) | | 18.5 (11.72) | | 33.68 (97.71) | |
| Facility complexity | | | <0.001 | | <0.001 | | <0.001 |
| Highest | 4065 (42.76%) | 11.69 (16.43) | | 50.33 (19.53) | | 67.38 (270.58) | |
| High | 2056 (21.63%) | 9.74 (12.94) | | 48.41 (24.22) | | 97.55 (469.68) | |
| Mid-High | 1598 (16.81%) | 8.11 (9.85) | | 28.49 (13.4) | | 35.65 (107.99) | |
| Medium | 843 (8.87%) | 5.66 (6.57) | | 20.06 (11.55) | | 32.85 (99.7) | |
| Low | 940 (9.89%) | 5.27 (5.29) | | 12.66 (8.46) | | 23.37 (65.31) | |
| Region | | | <0.001 | | <0.001 | | <0.001 |
| Continental | 1513 (16.1%) | 8.37 (12.47) | | 36.62 (21.16) | | 46.58 (181.03) | |
| Midwest | 2247 (23.91%) | 10.08 (13.48) | | 45.86 (27.41) | | 90.9 (442.24) | |
| North Atlantic | 2243 (23.86%) | 9.27 (13.83) | | 33.13 (20.87) | | 49.29 (242.24) | |
| Pacific | 1600 (17.02%) | 8.46 (11.32) | | 35.56 (18.34) | | 48.8 (181.79) | |
| Southeast | 1796 (19.11%) | 11.16 (15.29) | | 47.61 (21.3) | | 60.74 (245.74) | |

*sd, standard deviation; p-values resulting from one-way ANOVA; continuous variables categorized into quartiles

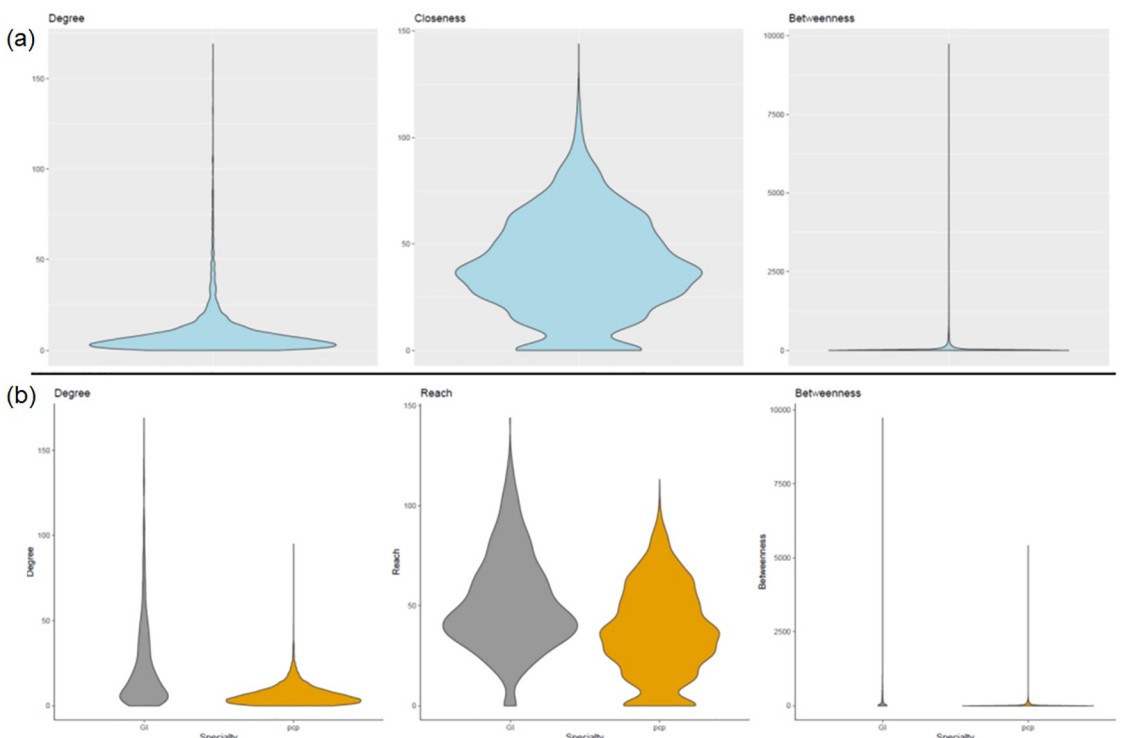

**Fig 2. Violin plot demonstrating the distribution of centrality measures (a) overall, (b) by specialty (GI, gastroenterology; PCP, primary care).**

12.66 [sd 8.46]), as they were more likely to have higher mean degree centrality and mean closeness centrality. However, clinicians affiliated with a high (mean betweenness centrality 97.55 [sd 469.68; p<0.001]) rather that the highest (mean betweenness centrality 67.38 [sd 270.58]) complexity designation had a higher mean betweenness centrality (Table 1).

In a weighted linear regression model adjusting for specialty, clinician type, practice location, facility complexity designation, and volume of patients with IBD, clinicians with a higher number of connections (degree centrality) were more likely to exhibit high-quality practices, with a lower percentage of patients who were prolonged steroid users (beta -0.0268; 95% CI -0.0427, -0.0110) and a higher percentage of prolonged steroid users receiving steroid-sparing therapy (beta 0.0967; 95% CI 0.0128, 0.1805). In contrast, clinicians who connect otherwise unconnected clinicians (betweenness) were more likely to exhibit low-

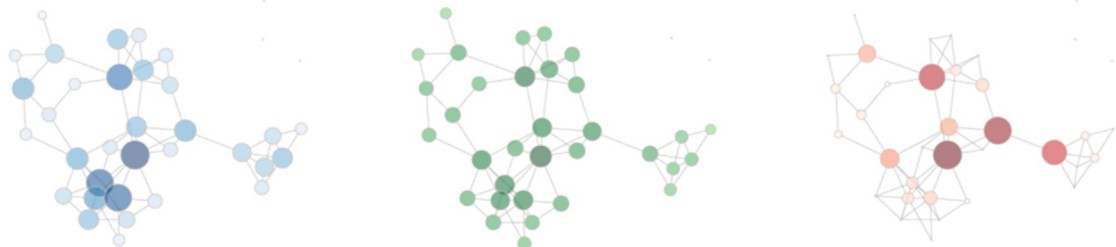

**Fig 3. Example of three centrality measures in one of the 130 facilities within the cohort, with darker nodes representing higher degree, closeness, and betweenness, respectively.**

**Table 2. Centrality and associated provider-level performance.**

| A. Unadjusted models | Three measures of provider centrality | | | | | | | | |
|---|---|---|---|---|---|---|---|---|---|
| **Performance Measure** | **Degree** | **95% CI** | **P-value** | **Closeness** | **95% CI** | **P-value** | **Betweenness** | **95% CI** | **P-value** |
| Prolonged steroid use | 0.0447 | 0.0257, 0.0636 | <0.001 | 0.0220 | 0.0110, 0.0330 | <0.001 | 0.0012 | 0.0003, 0.0021 | 0.007 |
| Use of immune targeted therapy in case of prolonged steroid use | 0.0947 | 0.0318, 0.1575 | 0.003 | 0.0619 | 0.0121, 0.1116 | 0.015 | 0.0033 | 0.0006, 0.006 | 0.018 |
| **B. Adjusted models*** | | | | | | | | | |
| **Performance Measure** | **Degree** | **95% CI** | **P-value** | **Closeness** | **95% CI** | **P-value** | **Betweenness** | **95% CI** | **P-value** |
| Prolonged steroid use | -0.0268 | -0.0427, -0.0110 | 0.001 | 0.0027 | -0.0067, 0.0121 | 0.571 | 0.0003 | 0.0001, 0.0006 | 0.012 |
| Use of immune targeted therapy in case of prolonged steroid use | 0.0967 | 0.0128, 0.1805 | 0.024 | -0.0244 | -0.0803, 0.0315 | 0.392 | 0.0011 | -0.0002, 0.0024 | 0.091 |
| **C. Subgroup analysis of gastroenterology providers**** | | | | | | | | | |
| **Performance Measure** | **Degree** | **95% CI** | **P-value** | **Closeness** | **95% CI** | **P-value** | **Betweenness** | **95% CI** | **P-value** |
| Prolonged steroid use | -0.0449 | -0.0669, -0.0229 | <0.001 | -0.0097 | -0.0262, 0.0067 | 0.246 | 0.0004 | 0.0000, 0.0007 | 0.032 |
| Use of immune targeted therapy in case of prolonged steroid use | 0.1196 | 0.0133, 0.2259 | 0.028 | -0.0033 | -0.0839, 0.0772 | 0.246 | 0.0013 | -0.0002, 0.0028 | 0.088 |

* adjusted for IBD volume, rurality, complexity, provider type, and specialty type

**GI providers only; adjusted for IBD volume, rurality, provider type, and complexity

quality practices, with a higher percentage of prolonged steroid users (beta 0.0003; 95% CI 0.0001,0.0006). A clinician's reach (closeness) was not associated with high-quality practices, including prolonged steroid use or use of steroid-sparing therapy (beta 0.0027; 95% CI -0.0067, 0.0121 and beta 0.0244; 95% CI -0.0803, 0.0315, respectively) (Table 2). To understand this in the context of clinical significance, we simulated how the relationship between degree centrality and the pre-specified primary endpoints might apply to clinicians with different levels of centrality (Fig 4). The presence of variation is more relevant than its magnitude or directionality.

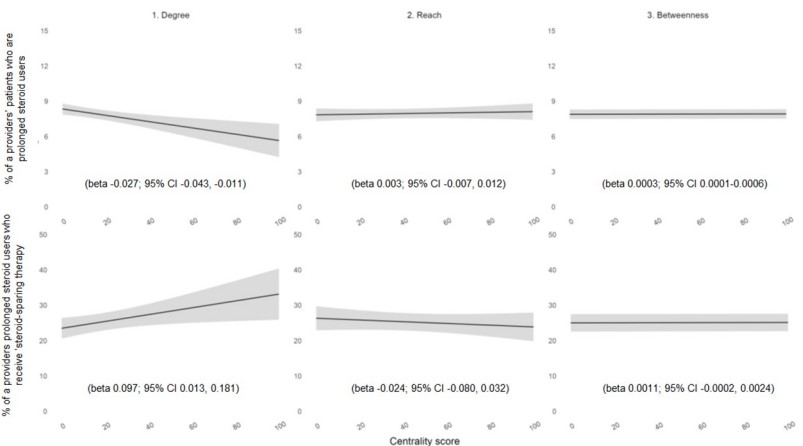

**Fig 4. Marginal effects plots representing the relationship between provider centrality (collaboration) and high-quality clinical practices.**

Given the different roles of gastroenterologists and primary care providers in IBD care, we conducted a sensitivity analysis only including gastroenterologists. Highly connected gastroenterologists have similar characteristics to the full cohort (S1 Table). After adjusting for clinician type, practice location, facility complexity designation, and volume of patients with IBD, we see that gastroenterologists with a higher number of connections (degree centrality) were more likely to exhibit high-quality clinical practices with a lower percentage of prolonged steroid users (beta -0.0449; 95% CI -0.0669, -0.0229) and a higher percentage of prolonged steroid users receiving steroid-sparing therapy with an immunomodulator or biologic (beta 0.1196; 95% CI 0.0133, 0.2259). Gastroenterologists who connected otherwise unconnected clinicians (betweenness) were more likely to exhibit low-quality practices with a higher percentage of prolonged steroid users (beta 0.0004; 95% CI 0.0000–0.0007). A gastroenterologist's reach (closeness) was not associated with the quality of clinical practice, whether prolonged steroid use or use of steroid-sparing therapy (beta -0.0097; 95% CI -0.0262, 0.0067 and beta -0.0033; 95% CI -0.0839, 0.0772, respectively) (Table 2).

## Discussion

Prior studies have shown us that designing interventions that target highly connected clinicians in a network has the potential to facilitate the dissemination of high-quality care and improve IBD outcomes. In a national cohort study and controlling for clinician characteristics, we found that clinicians with a high number of collaborators are more likely to exhibit high-quality IBD practices. Therefore, they may be well positioned to disseminate high-quality care such as minimizing the use of prolonged steroids and maximizing the use of steroid-sparing therapy to improve clinical outcomes. However, clinicians who connect clinicians who are otherwise unconnected are more likely to display low-quality IBD practices. These findings suggest that there is room for improving the practices of some highly connected clinicians prior to leveraging clinician networks to widely disseminate high-quality IBD practices.

Highly connected clinicians with a high degree, closeness, or betweenness are more likely to see a high volume of IBD patients and are more likely to be gastroenterologists, which is consistent with current referral patterns. This suggests that high volume clinicians and gastroenterologists may be best positioned to disseminate IBD clinical practices. However, access to IBD-specialized clinicians and high-volume IBD centers is limited. It is interesting that in this sample, physician assistants are highly connected compared to physicians, though they represent a minority of clinicians who care for patients with IBD. This may speak to the role physician assistants play in the current VA health system structure, with a greater role in specialty care as compared to primary care. Finally, it is important to note that highly connected clinicians are more likely to be in urban practices and are more likely to be affiliated with facilities with higher complexity designations than clinicians at rural practices and at facilities with lower complexity designations. While expected, this speaks to the challenges of disseminating high-quality IBD clinical practices to rural sites and lower complexity facilities [21, 22].

This study's strengths include a large sample size, availability of national multifacility data, and previously established methods for constructing a patient-sharing healthcare network. We also consider clinician connections in context of clinical practice-related factors. However, the limitations of the study need to be considered. First, our analysis was limited to clinicians in primary care and gastroenterology, as these are the two most common clinicians in an IBD patient's care team. However, this does not represent extended networks of surgeons, rheumatologists, or other specialists that some patients require. These larger

networks could be expanded upon in future work, depending on the outcome of interest and interventions considered. Second, considering a population management approach, we defined our outcome variables as a percentage of a clinician's patients who achieved the outcome of interest and all clinicians who cared for a patient were considered equally accountable for their care. Therefore, a patient with prolonged steroid use who is prescribed corticosteroids by several clinicians (e.g., gastroenterologist, emergency department physician) contributes to the percentage of prolonged steroid users for all the clinicians who are responsible for their care, regardless of who prescribed each steroid prescription. We chose this definition for our outcomes measure to capture low quality patient care across the inherently fragmented U.S. healthcare system, rather than to place blame on any individual clinician. In addition, no standard way for identifying accountable clinicians exists. Third, network characteristics such as the patient-sharing connections (edges) do not speak to direct interaction or communication between clinicians. Our analysis cannot determine directionality or causality, and so it is possible that that clinicians with high-quality IBD practices are more likely to get referrals, explaining their higher number of connections. Regardless, the study findings provide a foundation from which to design targeted interventions. Finally, we do not examine all IBD quality indicators including completing tuberculosis and hepatitis B screening prior to initiating an anti-tumor necrosis factor medication, adhering to colorectal cancer surveillance every 1–3 years, and receiving pneumococcal and influenza vaccinations while on immunosuppression. Future work should extend this study's findings to these other quality indicators.

Our overarching goal is to improve the quality of care and health outcomes for patients with IBD. Current efforts to improve care commonly focus on organizational redesign (e.g., IBD-specialized medical homes) as a strategic target. However, organizational redesign is not often feasible. This study is innovative in its effort to better understand existing clinician networks to inform the design of interventions that leverage existing networks to disseminate high-quality treatment. This study's findings provide insight into how clinician connectedness may be leveraged for future quality improvement interventions and dissemination efforts. The study specifically identifies the potential need for improving the practices of highly connected clinicians prior to investing in widespread efforts to disseminate high-quality IBD care. Network analysis methodology provides a foundation for understanding clinician connections as a means for improving high-quality care. An understanding of clinician networks, the variation in the level of connection among clinicians, and the characteristics of highly connected clinicians informs further strategies for improving IBD care through the dissemination of high-quality practices. Initial interventions may target highly connected clinicians through educational initiatives, academic detailing, and standardized care pathways, in addition to other established interventions for improving high-quality care. Depending on their impact, structuring IBD care around highly connected clinicians, through a centralized clinical hub or virtual IBD center could also be considered.

## Supporting information

**S1 Table. Measures of centrality and associated provider characteristics for GI providers only.**
(DOCX)

**S1 Checklist. STROBE statement—Checklist of items that should be included in reports of *cross-sectional studies*.**
(DOCX)

## Author Contributions

**Conceptualization:** Shirley Cohen-Mekelburg, Tony Van, Xianshi Yu, Deena Kelly Costa, Sameer Saini, Heather Gilmartin, Andrew J. Admon, Ken Resnicow, Peter D. R. Higgins, Geoffrey Siwo, Ji Zhu, Akbar K. Waljee.

**Data curation:** Tony Van, Xianshi Yu.

**Formal analysis:** Tony Van, Xianshi Yu.

**Funding acquisition:** Shirley Cohen-Mekelburg, Peter D. R. Higgins, Akbar K. Waljee.

**Investigation:** Shirley Cohen-Mekelburg, Tony Van, Xianshi Yu, Deena Kelly Costa, Milisa Manojlovich, Sameer Saini, Heather Gilmartin, Andrew J. Admon, Ken Resnicow, Peter D. R. Higgins, Geoffrey Siwo, Ji Zhu, Akbar K. Waljee.

**Methodology:** Shirley Cohen-Mekelburg, Tony Van, Xianshi Yu, Deena Kelly Costa, Milisa Manojlovich, Sameer Saini, Heather Gilmartin, Andrew J. Admon, Ken Resnicow, Peter D. R. Higgins, Geoffrey Siwo, Ji Zhu, Akbar K. Waljee.

**Resources:** Akbar K. Waljee.

**Supervision:** Shirley Cohen-Mekelburg, Peter D. R. Higgins, Akbar K. Waljee.

**Validation:** Shirley Cohen-Mekelburg.

**Visualization:** Shirley Cohen-Mekelburg, Tony Van, Xianshi Yu.

**Writing – original draft:** Shirley Cohen-Mekelburg, Tony Van, Sameer Saini, Heather Gilmartin, Andrew J. Admon, Peter D. R. Higgins, Akbar K. Waljee.

**Writing – review & editing:** Shirley Cohen-Mekelburg, Tony Van, Xianshi Yu, Deena Kelly Costa, Milisa Manojlovich, Sameer Saini, Heather Gilmartin, Andrew J. Admon, Ken Resnicow, Peter D. R. Higgins, Geoffrey Siwo, Ji Zhu, Akbar K. Waljee.

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
