## [Decision Letter · Decision Letter 0]

13 Oct 2022

PONE-D-22-22355Understanding Clinician Connections to Inform Efforts to Promote High-Quality Inflammatory Bowel Disease CarePLOS ONE

Dear Dr. Cohen-Mekelburg,

Thank you for submitting your manuscript to PLOS ONE. After careful consideration, we feel that it has merit but does not fully meet PLOS ONE’s publication criteria as it currently stands. Therefore, we invite you to submit a revised version of the manuscript that addresses the points raised during the review process.

We look forward to receiving your revised manuscript.

Kind regards,

Alpamys Issanov

Academic Editor

PLOS ONE

Journal Requirements:

Reviewers' comments:

Reviewer's Responses to Questions

**Comments to the Author**

1. Is the manuscript technically sound, and do the data support the conclusions?

Reviewer #1: Partly

Reviewer #2: Partly

2. Has the statistical analysis been performed appropriately and rigorously? 

Reviewer #1: I Don't Know

Reviewer #2: Yes

3. Have the authors made all data underlying the findings in their manuscript fully available?

Reviewer #1: No

Reviewer #2: No

4. Is the manuscript presented in an intelligible fashion and written in standard English?

Reviewer #1: Yes

Reviewer #2: Yes

5. Review Comments to the Author

Reviewer #1: Abstract

- ‘implement health practices’ – does this mean implement evidence based health practices? What kind of health practices? Who is the arbiter of the health practices that should be implemented?

- The logic of the 3rd sentence is not clear and the overall objectives of this research is not clear in the abstract.

- The concept of ‘unconnected clinicians’ is introduced in the abstract without definition. A corollary to this is that the concept of connected clinicians is also not clearly defined. This seems critical to your hypothesis. Suggest clarification.

- Why is ‘high quality practice’ defined by only one variable (‘lower use of prolonged steroids and higher use of steroid-sparing therapy’)? Is there evidence that the evidence based use of steroids is a surrogate for ‘high quality practice’? How is high quality practice defined?

- The numerical results in the abstract are not intuitively interpreted. It is difficult to understand how these estimates serve to support the study’s hypothesis. Could these estimates be converted into a scale that would be more intuitively interpreted?

- The last sentence does not necessarily follow from your data. Your findings appear to demonstrate that doctors with many connections (with other ‘connected’ physicians?) tend to use steroids appropriately. It is not clear that this ‘provides insight’ and it is not clear what is meant by ‘leverage’.

- “are well positioned to disseminate high quality care” It is not clear what this statement means (how does one disseminate ‘care’ ? should it be dissemination of knowledge about how high quality care can be provided?) and it is not clear how this statement flows from the results

Background

- “These negative health outcomes…” please provide support for this statement

- “For example, we know…” – prolonged steroid use could be due to many factors unrelated to dissemination of information (e.g. cost of therapy, patient willingness to use biologic agents)

- Support for one of the foundational hypothesis comes from a Honduran study, whose external validity may not include patients in America. Suggest referencing data that are generalizable to the population you are studying.

- ‘This gastroenterologist…’ – has it been demonstrated that this is the way that evidence-based practice measures are introduced into a clinical community?

- Last sentence – it needs to be clearer why you need to ‘improve the practices’ (and how/if this can actually be accomplished) of the physicians that you identify, who already have good practices (according to your steroid surrogate measure). If they have good practices, why do they need to be improved?

- “Rather we aimed to characterize….” “our objective is not to examine…” - this is confusing because your results in the abstract appear to be focused on expressing statistical association between connectedness and appropriate steroid usage, your surrogate for good clinical practice. This is different from ‘characterizing’.

Discussion

- “therefore we must leverage the health system…” this claim does not seem to follow from the prior sentences

- Concepts seem to be repeated in this section. Suggest abbreviating the Discussion.

-

Reviewer #2: Given study aims to identify the potential need for improving the practices of highly connected clinicians prior to investing in

widespread efforts to disseminate high-quality IBD clinical practices. Although, the investigation seems to address important issue, revisions are required.

1) Consider rephrasing the following sentence:

"A randomized controlled trial evaluated the adoption of multivitamins to prevent and treat micronutrient deficiencies across remote Honduran villages and compared a strategy of targeting the intervention at highly connected individuals nominated by others as compared to a non-targeted strategy."

2) Authors aims and objectives should have been defined clearly. Since the quality of treatment is assessed and analyzed, researchers are evaluating the association between these variables. The purpose of stating that they did not aim to examine causal relationship and reasons are not clear.

3) Did authors use the standardized criteria to evaluate IBD using ICD-10 codes for ulcerative colitis and Crohn’s disease? Are these codes exhaustive? Could authors clarify on validity of these ICD-10 codes? Since there were no other instruments or lab test results, it is important for ICD-10 code to properly identify the sample.

4) Were there any other inclusion or exclusion criteria used in this study?

5) Study outcomes need to be explained with more details.

6) In table, the authors indicate some p-values as "0.0000". It is better to indicate that is it less that some value, if it very small p-value, for example, "<0.0001". The same is true for the limits of confidence intervals. Consider rounding the boundaries of confidence intervals indicated as "0.0000".

7) Consider rephrasing the sentence "However, connected clinicians who connect otherwise unconnected clinicians are more likely to display low-quality IBD practices".

8) Discussion part seems to be too long. Consider being consive in discussing the findings of the study.

6. PLOS authors have the option to publish the peer review history of their article (what does this mean?). If published, this will include your full peer review and any attached files.

Reviewer #1: No

Reviewer #2: No

---

## [Author Response · Author response to Decision Letter 0]

20 Oct 2022

Dear Editors,

We thank you for taking the time to review and provide thoughtful feedback on our manuscript submission titled “Understanding Clinician Connections to Inform Efforts to Promote High-Quality Inflammatory Bowel Disease Care” in PLOS ONE.

We found the comments and suggestions of the editorial committee and reviewers to be extremely helpful in improving the quality of the manuscript. We believe that we have addressed the points that were raised and have provided a detailed description of changes to the manuscript. Below you will find a point-by-point response to reviewer comments along with two attached revised drafts – a clean revised manuscript and a second revised manuscript with any changes marked in red in Microsoft Word. 

To summarize, we have made major revisions to our Abstract and Background to clarify the study objectives, define connections, and the selection of specific IBD quality indicators. We also better describe social network theory to support our hypothesis and the concept of network connections. We have also clarified our inclusions/exclusions and provide references to support how we identified patients using ICD-10 codes. Finally, we have clarified and shortened our Discussion to emphasize key points and minimize redundancy.

Thank you for your consideration of our revised manuscript. We look forward to your response.

Sincerely, 

Shirley Cohen-Mekelburg, M.D., M.S. 

Assistant Professor

Division of Gastroenterology & Hepatology

Department of Internal Medicine

University of Michigan

Point by Point Responses:

Reviewer #1: 

We appreciate your review and feedback.

Abstract:

1. ‘Implement health practices’ – does this mean implement evidence-based health practices? What kind of health practices? Who is the arbiter of the health practices that should be implemented? The logic of the 3rd sentence is not clear and the overall objective of this research is not clear in the abstract.

Thank you for this feedback. We have revised the background section of the Abstract to clarify. We now state, “Highly connected individuals disseminate information effectively within their social network. To apply this concept to inflammatory bowel disease (IBD) care and lay the foundation for network interventions to disseminate high-quality treatment, we assessed the need for improving the IBD practices of highly connected clinicians. We aimed to examine whether highly connected clinicians who treat IBD patients were more likely to provide high-quality treatment than less connected clinicians.” (Page 2)

2. The concept of ‘unconnected clinicians’ is introduced in the abstract without definition. A corollary to this is that the concept of connected clinicians is also not clearly defined. This seems critical to your hypothesis. Suggest clarification.

The definition of clinician connections in this study is based on patient-sharing between clinicians which is an established definition that has been described in prior literature on network analysis in healthcare. We have clarified the definition of clinician connections and now state, “We created a network comprised of clinicians connected by shared patients. We quantified clinician connections using degree centrality (number of clinicians with whom a clinician shares patients), closeness centrality (reach via shared contacts to other clinicians), and betweenness centrality (degree to which a clinician connects clinicians not otherwise connected).” (Page 2) 

3. Why is ‘high quality practice’ defined by only one variable (‘lower use of prolonged steroids and higher use of steroid-sparing therapy’)? Is there evidence that the evidence-based use of steroids is a surrogate for ‘high quality practice’? How is high quality practice defined?

Minimizing the use of prolonged steroids and maximizing the use of steroid-sparing therapy are two IBD quality indicators described by the Crohn’s Colitis Foundation and American Gastroenterological Association (See Reference #9 and #16). Prior data suggests that there is wide variation in the use of prolonged steroids and the use of steroid-sparing therapies (see References #5, 17, 18). These measures directly relate to the ability to achieve steroid-free remission and prevent IBD-related complications, which is why we chose them as our outcome measures. Other IBD quality indicators include completing tuberculosis and hepatitis B screening prior to initiating anti-tumor necrosis factor medications, adhering to colorectal cancer surveillance every 1-3 years, and receiving pneumococcal and influenza vaccinations while on immunosuppression. Many of these other indications are more difficult to capture without high rates of missingness using electronic health data. Future work should extend this study’s findings to these other quality indicators. We clarified that these are two of several IBD quality indicators in our Abstract Methods section, stating “Using weighted linear regression, we examined associations between each measure of connection and two IBD quality indicators: low prolonged steroids use, and high steroid-sparing therapy use.” (Page 2) We have also added the rationale for selecting these measures in the Methods section of the manuscript (Page 7): “There were two primary outcomes representative of high-quality IBD clinical practices: clinicians with fewer patients who are prolonged steroid users and clinicians with a higher percentage of previous or current prolonged steroids users receiving steroid-sparing therapy. Both outcomes are considered quality metrics in IBD care and recommendations are to minimize use of prolonged steroids and maximize steroid-sparing therapy with immunomodulators or biologics in this context, though threshold benchmarks do not currently exist.9,16 Prior data suggests that there is wide variation in the use of prolonged steroids and the use of steroid-sparing therapies.5,17,18 These measures directly relate to the ability to achieve steroid-free remission and prevents IBD-related complications, which is why we chose them as our outcome measures.” Finally, we have added mention in the limitations portion of the Discussion section that we do not examine other IBD quality indicators, and that future work should do so. (Page 16)

4. The numerical results in the abstract are not intuitively interpreted. It is difficult to understand how these estimates serve to support the study’s hypothesis. Could these estimates be converted into a scale that would be more intuitively interpreted? 

We have included Figure 4 in the manuscript to help with this interpretation. In order to further facilitate the overarching interpretation in the results section of the abstract, we have added clarification and state, “The presence of variation is more relevant than its magnitude.” (Page 2)

5. The last sentence does not necessarily follow from your data. Your findings appear to demonstrate that doctors with many connections (with other ‘connected’ physicians?) tend to use steroids appropriately. It is not clear that this ‘provides insight’ and it is not clear what is meant by ‘leverage’. “are well positioned to disseminate high quality care” It is not clear what this statement means (how does one disseminate ‘care’ ? should it be dissemination of knowledge about how high quality care can be provided?) and it is not clear how this statement flows from the results.

We have modified this section to more clearly describe the interpretation of our study findings and how they inform the overarching goal of developing interventions that leverage the clinician network to disseminate high quality IBD practices. The Abstract conclusion section now states, “Clinicians with a high number of connections provided more high-quality IBD treatments than less connected clinicians and may be well-positioned for interventions to disseminate high-quality IBD care. However, clinicians who connect clinicians who are otherwise unconnected are more likely to display low-quality IBD practices. Efforts to improve their quality are needed prior to leveraging their position to disseminate high-quality practices.” (Page 3)

Background

6. “These negative health outcomes…” please provide support for this statement. “For example, we know…” – prolonged steroid use could be due to many factors unrelated to dissemination of information (e.g., cost of therapy, patient willingness to use biologic agents)

Our intent in this statement is to introduce the concept that variation in health outcomes in IBD, as in other chronic diseases, is due in part to how care is delivered. There are multilevel factors that contribute to this variation, including cost of therapy and patient preferences, and variation in prolonged steroids and steroid-sparing therapy are two examples of this variation. We have clarified this concept by revising paragraph 1 in the Background to state, “These negative health outcomes result in part from variation in clinical practice, which are influenced by patient, clinician, and organizational factors. One example of this variation in IBD care is the common use of prolonged steroid use in some facilities and by some clinicians, and the underutilization of steroid-sparing therapies.5,6 There is an urgent need to understand how to better optimize the dissemination and adoption of high-quality clinical practices to minimize variation in high-quality clinical practice and deliver better IBD care.” (Page 4)

7. Support for one of the foundational hypotheses comes from a Honduran study, whose external validity may not include patients in America. Suggest referencing data that are generalizable to the population you are studying.

The concept of designing interventions that target particular individuals within a network to disseminate information and behavioral change is relatively novel. The interventional study described in the Background by Kim et al. is a classic example how network interventions can be applied to improve health. This example focuses on disseminating public health practices within a Honduran social network. We are not aware of similar network interventions that have been evaluated within the U.S. healthcare system. Our overarching goal is to apply these innovative concepts towards developing a similar intervention in the U.S. health care system. This study is a first step towards achieving that goal. Therefore, we clarify in the Background (page 4), “The study found that the strategy of targeting highly connected individuals was more effective in disseminating high-quality multivitamin practices. Our overarching goal is to apply this knowledge from the public health literature to facilitate the dissemination of high-quality clinical practices among clinicians in a health system network.”

8. ‘This gastroenterologist…’ – has it been demonstrated that this is the way that evidence-based practice measures are introduced into a clinical community? 

This description is supported by social network theory, which has been applied to a wide variety of health areas. We now introduce this concept in more detail in the Background, and state, “Collaboration and exchange of information between individuals facilitate the spread of knowledge and adoption of beneficial practices. This is a central tenet of social network theory, which emphasizes the importance of connections. In social network theory, individuals take action based on their network’s environment and an individual’s positioning in a network influences their behavior.7 These concepts have been applied to a wide variety of health areas.” (Page 5)

9. Last sentence – it needs to be clearer why you need to ‘improve the practices’ (and how/if this can actually be accomplished) of the physicians that you identify, who already have good practices (according to your steroid surrogate measure). If they have good practices, why do they need to be improved?

Variation in IBD care is common as is described in paragraph 1 of the Background section. Our overarching goal is to reduce this variation by designing interventions to improve dissemination of high quality IBD practices using a social network approach. As a first step, we need to understand variation in IBD clinical practices among clinicians who are highly connected and those who are not, to identify if there is a need to first focus on improving the clinical practices of highly connected individuals before focusing on effective dissemination of these practices using a network approach. To clarify these points, we now state that “This study will identify the potential need for improving the practices of highly connected clinicians, laying the foundation for developing interventions that leverage clinician networks to disseminate high-quality IBD care. (Page 6)

10. “Rather we aimed to characterize….” “our objective is not to examine…” - this is confusing because your results in the abstract appear to be focused on expressing statistical association between connectedness and appropriate steroid usage, your surrogate for good clinical practice. This is different from ‘characterizing’.

We agree. We have modified our study objective. We now state, ““our study objective is to examine whether highly connected clinicians who treat patients with IBD are more likely to provide high-quality IBD care than less connected clinicians.” (Page 6)

Discussion

11. “Therefore, we must leverage the health system…” this claim does not seem to follow from the prior sentences

In response to #12 below, we have removed this sentence. 

12. Concepts seem to be repeated in this section. Suggest abbreviating the Discussion.

We have revised the Discussion section to be more concise and avoid redundancy (Pages 15-18).

Reviewer #2: Given study aims to identify the potential need for improving the practices of highly connected clinicians prior to investing in widespread efforts to disseminate high-quality IBD clinical practices. Although, the investigation seems to address important issue, revisions are required.

We appreciate your review and feedback.

13. Consider rephrasing the following sentence: "A randomized controlled trial evaluated the adoption of multivitamins to prevent and treat micronutrient deficiencies across remote Honduran villages and compared a strategy of targeting the intervention at highly connected individuals nominated by others as compared to a non-targeted strategy."

We have revised this sentence for clarity. It now reads, “One classic example is that of a randomized controlled trial across Honduran villages that compared a strategy of targeting a group of public health interventions at highly connected individuals nominated by others as compared to a non-targeted strategy. The interventions included multivitamins for micronutrient deficiencies, and chlorine for water purification. The study found that the strategy of targeting highly connected individuals was more effective in disseminating these high-quality public health practices.” (Page 4) 

14. Authors’ aim and objectives should be defined clearly. Since the quality of treatment is assessed and analyzed, researchers are evaluating the association between these variables. The purpose of stating that they did not aim to examine causal relationship and reasons are not clear.

We have revised this paragraph to remove the statement describing how we are not examining causal relationships. Rather, we state, “Our study objective is to examine whether highly connected clinicians who treat patients with IBD are more likely to provide high-quality IBD care than less connected clinicians.” (Page 6)

15. Did authors use the standardized criteria to evaluate IBD using ICD-10 codes for ulcerative colitis and Crohn’s disease? Are these codes exhaustive? Could authors clarify on validity of these ICD-10 codes? Since there were no other instruments or lab test results, it is important for ICD-10 code to properly identify the sample.

Yes, patients with IBD were identified using a combination of International Classification of Diseases, Tenth Revision (ICD-10) codes for ulcerative colitis and Crohn’s disease that have been validated in VHA data. We use the same ICD-10 codes as Khan et al. who validated these ICD-10 codes in 2018. They manually reviewed the VHA records of 200 randomly selected patients who had any of the IBD diagnostic codes and confirmed accuracy with a true IBD diagnosis with 94.5% precision and 100% concordance between 2 reviewers. We reference this study (#15) in the Methods section (Page 6) (https://pubmed.ncbi.nlm.nih.gov/29309905/)

16. Were there any other inclusion or exclusion criteria used in this study?

We included all patients who carried an IBD diagnosis and were managed within the VHA between 2015 and 2018. There were no additional exclusions. The clinicians in this study that were included were physicians, nurse practitioners, and physician assistants who cared for patients with IBD via office visit encounters. All gastroenterology providers and primary care providers who cared for an IBD patient through at least one office visit were included. Other specialties were excluded as they are less likely to be accountable for chronic IBD care. We have revised the location of this information within the Methods section to make this clearer. (Page 6-7)

17. Study outcomes need to be explained with more details.

We have added details to the results where they were previously deficient. For example, on Page 12-13, we now state, “Gastroenterologists were more likely to be highly connected than primary care providers, with a higher mean degree centrality (gastroenterologist 23.19 [sd 25.48] vs. primary care providers 6.60 [sd 5.76], p < 0.001), mean closeness centrality (gastroenterologist 51.42 [sd 23.87] vs. primary care providers 37.37 [sd 22.13], p<0.001), and mean betweenness centrality (gastroenterologist 230.79 [sd 616.55] vs. primary care providers 25.36 [sd 113.65], p<0.001). Physician assistants were similarly more likely to be highly connected (mean degree 10.75 [sd 15.11] p=0.038, mean closeness 40.70 [sd 24.60] p=0.002, and mean betweenness 85.16 [sd 351.05] p=0.076) than physicians (mean degree 9.38 [sd 13.52], mean closeness 40.18 [sd 22.98], and mean betweenness 59.27 [sd 300.98]) or nurse practitioners (mean degree 9.43 [sd 12.36], mean closeness 38.18 [sd 22.74], and mean betweenness 59.18 [sd 203.65]) (Table 1). Clinicians with a higher IBD volume and more IBD visit encounters were also more connected than those with lower IBD volumes and fewer visit encounters. Clinicians with the top quartile of IBD patient volume demonstrated a mean degree centrality of 21.88 (sd 21.43, p<0.001), mean closeness centrality of 52.40 (sd 23.57, p<0.001), and mean betweenness centrality of 193.79 (sd 541.20, p<0.001) as compared to clinicians at the bottom quartile of IBD patient volume (mean degree centrality 2.55 (sd 2.37), mean closeness centrality 28.54 (21.22), and mean betweenness centrality 3.73 (sd 21.09). Clinicians with the top quartile of IBD visit volume demonstrated a mean degree centrality of 20.01 (sd 21.97, p<0.001), mean closeness centrality of 50.33 (sd 24.07, p<0.001), and mean betweenness centrality of 180.54 (sd 537.83, p<0.001) as compared to clinicians at the bottom quartile of IBD patient volume (mean degree centrality 2.74 (sd 2.60), mean closeness centrality 28.82 (21.15), and mean betweenness centrality 4.39 (sd 23.12).” 

“Clinicians’ practice settings were also associated with their level of connection. Clinicians who practiced in an urban setting were more connected (mean degree centrality 9.83 [sd 13.84, p<0.001], mean closeness centrality 41.63 [sd 22.95, p<0.001], and mean betweenness centrality 63.09 [sd 298.82, p=0.033]) than those who practiced in a rural setting (mean degree centrality 5.91 [sd 6.54], mean closeness centrality 18.50 [sd 11.72], mean betweenness centrality 33.68 [sd 97.71]). Finally, clinicians who were affiliated with facilities with the highest complexity designation were more connected (mean degree centrality 11.69 [sd 16.43, p<0.001], mean closeness centrality 50.33 [sd 19.53, p<0.001]) than those affiliated with facilities with the lowest complexity designation (mean degree centrality 5.27 [sd 5.29], mean closeness centrality 12.66 [sd 8.46]), as they were more likely to have higher mean degree centrality and mean closeness centrality. However, clinicians affiliated with a high (mean betweenness centrality 97.55 [sd 469.68, p<0.001]) rather that the highest (mean betweenness centrality 67.38 [sd 270.58]) complexity designation had a higher mean betweenness centrality (Table 1).”

18. In table, the authors indicate some p-values as "0.0000". It is better to indicate that is it less that some value, if it very small p-value, for example, "<0.0001". The same is true for the limits of confidence intervals. Consider rounding the boundaries of confidence intervals indicated as "0.0000".

In response, we have rounded p-values in all the Tables to the 3rd decimal. By rounding the confidence intervals we would lose some degree of meaning in the study findings as so we have kept confidence interval to the 4th decimal place.

19. Consider rephrasing the sentence “However, connected clinicians who connect otherwise unconnected clinicians are more likely to display low-quality IBD practices”.

We have revised this to say, “clinicians who connect clinicians who are otherwise unconnected are more likely to display low-quality IBD practices.” (Page 14)

20. Discussion part seems to be too long. Consider being concise in discussing the findings of the study.

As described above in Response #12, we have revised the Discussion section to be more concise (Pages 15-18).

---

## [Decision Letter · Decision Letter 1]

7 Dec 2022

Understanding Clinician Connections to Inform Efforts to Promote High-Quality Inflammatory Bowel Disease Care

PONE-D-22-22355R1

Dear Dr. Cohen-Mekelburg,

We’re pleased to inform you that your manuscript has been judged scientifically suitable for publication and will be formally accepted for publication once it meets all outstanding technical requirements.

Kind regards,

Alpamys Issanov

Academic Editor

PLOS ONE

Additional Editor Comments (optional):

Reviewers' comments:

Reviewer's Responses to Questions

**Comments to the Author**

1. If the authors have adequately addressed your comments raised in a previous round of review and you feel that this manuscript is now acceptable for publication, you may indicate that here to bypass the “Comments to the Author” section, enter your conflict of interest statement in the “Confidential to Editor” section, and submit your "Accept" recommendation.

Reviewer #1: All comments have been addressed

Reviewer #2: All comments have been addressed

2. Is the manuscript technically sound, and do the data support the conclusions?

Reviewer #1: Yes

Reviewer #2: Yes

3. Has the statistical analysis been performed appropriately and rigorously? 

Reviewer #1: I Don't Know

Reviewer #2: Yes

4. Have the authors made all data underlying the findings in their manuscript fully available?

Reviewer #1: No

Reviewer #2: No

5. Is the manuscript presented in an intelligible fashion and written in standard English?

Reviewer #1: Yes

Reviewer #2: Yes

6. Review Comments to the Author

Reviewer #1: Thanks for your responses to my comments. I still suggest rephrasing "intervention to disseminate..." The verb disseminate usually requires the subject of the sentence to be animate; the object of this verb is usually 'information'. It's a grammatical point, but because the meaning of this phrase is critical to the message of the paper, strongly suggest rephrasing this throughout. People disseminate knowledge. Interventions do not disseminate. In normative English, we do not 'disseminate' care, we disseminate information.

Your use of the word 'leverage' is not clear in meaning. Suggest deleting this jargon which does not help to communicate the implications of your study.

Reviewer #2: No further comments from my side

7. PLOS authors have the option to publish the peer review history of their article (what does this mean?). If published, this will include your full peer review and any attached files.

Reviewer #1: No

Reviewer #2: No

---

## [Editor Report · Acceptance letter]

12 Dec 2022

PONE-D-22-22355R1 

Understanding Clinician Connections to Inform Efforts to Promote High-Quality Inflammatory Bowel Disease Care 

Dear Dr. Cohen-Mekelburg:

I'm pleased to inform you that your manuscript has been deemed suitable for publication in PLOS ONE. Congratulations! Your manuscript is now with our production department. 

Kind regards, 

on behalf of

Dr. Alpamys Issanov 

Academic Editor

PLOS ONE